# Can routine assessment of older people's mental health lead to improved outcomes: A regression discontinuity analysis

**Kalpita Baird**[1]*, **Ailish Byrne**[1], **Sarah Cockayne**[1], **Rachel Cunningham-Burley**[1], **Caroline Fairhurst**[1], **Joy Adamson**[1], **Wesley Vernon**[2], **David J. Torgerson**[1], on behalf of the REFORM trial[¶]

**1** York Trials Unit, Department of Health Sciences, University of York, York, United Kingdom, **2** University of Huddersfield, Huddersfield, United Kingdom

¶ A full membership list of the author group can be found in the Acknowledgments.
* kalpita.baird@york.ac.uk

**Data Availability Statement:** "There are ethical restrictions on sharing a de-identified data set; consent to share anonymous data from the study

## Abstract

### Objective

To assess whether case finding for depression among people aged 65 and above improves mental health.

### Design

Opportunistic evaluation using a regression discontinuity analysis with data from a randomised controlled trial.

### Setting

The REFORM trial, a falls prevention study that recruited patients from NHS podiatry clinics.

### Participants

1010 community-dwelling adults over the age of 65 with at least one risk factor for falling (recent previous fall or fear of falling).

### Intervention

Letter sent to patient's General Practitioner if they scored 10 points or above on the 15-item Geriatric Depression Scale (GDS-15) informing them of the patient's risk of depression.

### Main outcome measure

GDS-15 score six months after initial completion of GDS-15.

### Results

895 (88.6%) of the 1010 participants randomised into REFORM had a valid baseline and six-month GDS-15 score and were included in this study. The mean GDS-15 baseline score was 3.5 (SD 3.0, median 3.0, range 0–15); 639 (71.4%) scored 0–4, 204 (22.8%) scored

was not obtained from participants. Data are available upon reasonable request. Data requests, with statistical analysis plans, can be sent to the Department of Health Sciences (DOHS), University of York [dohs-stats@york.ac.uk] or to the corresponding author, and will be considered on a case-by-case basis. All data requests will be managed in accordance with York Trials Unit, University of York, standard operating procedures."

**Funding:** The author(s)s received no specific funding for this work.

**Competing interests:** I have read the journal's policy and the authors of this manuscript have the following competing interests: SC, CF, JA, WV and DT received funding from the National Institute for Health and Care Research (NIHR, previously just called the National Institute for Health Research) Health Technology Assessment (HTA) paid to their employer, University of York, for the REFORM trial. All other authors declare no competing interests. This does not alter our adherence to PLOS ONE policies on sharing data and materials

5–9 indicating mild depression, and 52 (5.8%) scored 10 or higher indicating severe depression. At six months follow-up, those scoring 10 points or higher at baseline had, on average, a reduction of 1.08 points on the GDS-15 scale (95% confidence interval -1.83 to -0.33, p = 0.005) compared to those scoring less than 10, using the simplest linear regression model.

## Conclusion

Case finding of depression in podiatry patients based on a GDS-15 score of 10 or more followed by a letter to their General Practitioner significantly reduced depression severity. Whether this applies to all older patients in primary care is unknown. Further research is required to confirm these findings. Regression discontinuity analyses could be prespecified and embedded within other existing research studies.

## Introduction

Depression is the most prevalent mental health disorder amongst the older population [1]. It is estimated that one in four adults over the age of 65 years suffer from depression [2] but approximately 85% of those suffering do not receive help through the National Health Service (NHS) to manage their condition [3]. This may partially reflect patients' reluctance to disclose depressive symptoms to healthcare providers–potentially linked to a fear of the stigma surrounding mental health disorders [3]. An additional concern is evidence that General Practitioners (GPs) and those in first line contact with this population are faced with difficulties in identifying depressed patients [4, 5], in part due to the symptoms of depression amongst older adults varying considerably from those displayed by younger adults [6]. This lack of recognition likely contributes to a significant number of depressed patients failing to receive the help they need from healthcare services.

Whilst the United States Preventative Task Force recommends screening for depression in the general adult population [7], routine screening is not currently recommended in the United Kingdom (UK). Rather practitioners are advised to 'be alert' to possible depression (particularly in people with a history of depression or a chronic physical health problem) [8]. Furthermore, a large cluster randomised trial evaluating the screening for depression in primary care patients consulting with osteoarthritis found no evidence of a benefit [9].

The benefits of case finding for depression within primary care settings is widely debated [10–12]. There is limited evidence to suggest that patients screened for depression had better outcomes than patients who were not, when the same treatment resources were available to both groups [10]. Whilst screening could potentially identify people who do not recognise that they are experience depressive symptoms, leading to earlier detection, currently there is no strong evidence for routine case finding of depression among the older population [13]. In this paper we report a natural experiment using a regression discontinuity analysis to assess whether case finding of depression in older adults reduces depressive symptoms six months later.

## Materials and methods

This study was an unexpected evaluation resulting from the REFORM randomised controlled trial (RCT) for the prevention of falls. REFORM was a multicentre, two-armed RCT that recruited 1010 participants aged 65 and over with a risk factor for falling (recent previous fall

or fear of falling) to assess the effectiveness of a podiatry intervention for the prevention of falls in older people [14]. Regulatory approval for the study was obtained from the East of England–Cambridge East Research Ethics Committee (REC) on 9 November 2011 (REC reference number 11/EE/0379). Galway REC approved the study on 26 April 2013 (REC reference number C.A 886). The University of York, Department of Health Sciences Research Governance Committee approved the study on 2 August 2011. Research management and governance approval was obtained for each trust. Participation in the study was voluntary, and all participants gave written informed consent before taking part in the REFORM study. Patients were recruited from podiatry clinic lists and baseline data were collected from them prior to randomisation. Baseline data were collected between November 2012 and November 2014. The primary outcome was the incidence rate of falls over 12 months; however, secondary outcomes included the proportion of participants presenting with depression at six and 12 months post-randomisation. Within the REFORM trial, depression was classified through the use of the 15-item Geriatric Depression Scale (GDS-15) [15] measured at baseline, and again at six and 12 months post-randomisation. The GDS-15 (see **S1 Fig**) is a popular tool specifically targeted towards detecting depression amongst the older population and is a shortened version of the original 30-item scale [16, 17]. The scale includes items assessing how the responder has felt over the last week, including 'Are you basically satisfied with your life?' and 'Do you feel full of energy?'. A total score of 0–4 is considered normal, 5–9 indicates mild depression, and 10–15 indicates severe depression. The validity of the GDS-15 to identify depression within older cohorts, both in its original 30-item and shortened versions, within the UK and cross-culturally, has been displayed consistently across the literature [18–22]. The GDS-15 displays superior or equal levels of specificity, sensitivity, and appropriateness as several other popular depression scales, including the Beck Depression Inventory, Centre for Epidemiologic Studies Depression Scale, Zung Self-rating Depression Scale, and the Hamilton Rating Scale for Depression [23–27].

Participants in the REFORM trial completed and returned postal questionnaires to the York Trials Unit to collect participant-reported outcomes, including the GDS-15, at baseline, six and 12 months post-randomisation. On return of the questionnaire, a participant's GDS-15 score was calculated by hand by a trial coordinator. If a participant scored 10 or more, their consent form was reviewed to confirm whether they had given permission for the study team to contact their GP if any concerns were raised about their health during the course of the study. For those who agreed, an ethically approved letter was sent by second-class post to the participant's GP, detailing the participant's GDS-15 score, the date the scale was completed, and information about how to interpret the score. Based on this information, the letter recommended that a consultation with the participant be arranged as soon as practically possible to discuss possible treatment options for their depression that may be available, should the participant choose to take up any help. The trial manager's contact details and web link to the study protocol were provided in case the GP had any queries. This happened at each timepoint, except if the GP had been previously notified about a high GDS-15 score for a participant at an earlier trial assessment time point, they were not contacted again, even if the GDS-15 score was 10 or above on a subsequent assessment. We did not follow up with the patients or GPs regarding what, if any, action or treatment was undertaken as a result of the GP receiving our letter, as this was not related to any objectives of the main REFORM trial.

### Study design and statistical methods–Regression discontinuity design

A regression discontinuity design (RDD) is where a score or threshold is given to a continuous assignment variable and used to determine whether an individual is offered an intervention or

not [28, 29]. The continuous assignment variable used for this RDD was the participant's score at baseline on the GDS-15, where the threshold to receive the intervention (GP contacted) was a score of 10 (or more). The RDD considers the change in score six months later. No other interventions were applied on the basis of the participant's baseline GDS-15 score.

Two versions of RDD exist: 'sharp' and 'fuzzy'. Sharp RDD is deterministic in nature, meaning that all who meet the required threshold or score on the continuous assignment variable are offered the intervention (and those who do not meet the threshold are not); whereas fuzzy RDD is probabilistic, meaning that those who meet the required threshold or score are more likely to be offered the intervention than those who do not [30]. This study used a sharp RDD as all bar one participant who met the required threshold received the intervention (51/52, 98.1%). For the remaining participant, a letter was not sent to their GP as they did not consent to this. Given this only applies to one participant, this participant is included in all statistical analyses assuming that they did receive the intervention.

A limitation of the RDD is a concern around manipulation of the continuous assignment variable e.g. completion of the measure in such a way as to ensure the cut-point is met or just missed, depending on the context [31]. This is of limited concern here as the GDS-15 is a patient-reported outcome measure and participants were not aware, prior to its completion, that their scoring over a particular threshold on the GDS-15 would trigger a notification letter to their GP with recommendations for a follow-up visit. Manipulation of the assignment variable would therefore be unlikely. However, this was assessed visually using a histogram to check for clustering around the cut-point, which might indicate deliberate attempts to reach, or narrowly avoid, the threshold.

In order for the RDD to be valid, certain assumptions must be met [28, 30]. One assumption is that there is continuity in other variables between those who scored slightly above and those who scored slightly below the threshold [30]. This assumption would indicate that participants around the threshold are mainly similar in most other aspects except for the score on the assignment variable and, if met, would allow the estimation of causal effects of the intervention. Continuity amongst other variables around the cut-point was assessed using the methods described by Sood et al. 2014 [32]. The observed characteristics of participants that scored eight or nine and those that scored 10 or 11 (i.e. those falling just either side of the threshold) were compared, using a t-test or chi-squared test, as appropriate, at the 5% significance level.

To conduct the RDD, we considered a parametric approach [31], which uses every observation in the sample. Different functional forms of the regression model were estimated, increasing in complexity: linear, quadratic, cubic, all with and without interaction terms of the GDS-15 at baseline with treatment, and with and without adjusting for further covariates (participant's age, gender, REFORM trial allocation, and number of prescribed medications at baseline) to improve precision.

The simplest form, the linear regression model without an interaction, is:

$$y_i = \alpha_0 + \alpha_1 Above_i + \alpha_2 BaselineGDS_i + \varepsilon_i \tag{1}$$

Where:

- $y_i$ is participant $i$'s score on the GDS-15 at six months

- $\alpha_0$ is a constant

- $\varepsilon_i$ is a random error term

- $Above_i$ is an indicator variable that is equal to one if participant $i$'s score on the GDS-15 at baseline is 10 or more, and zero otherwise

- *BaselineGDS$_i$* is participant *i*'s score on the GDS-15 at baseline centred at the cut-point of 10.

- $\alpha_1$, $\alpha_2$ are the regression coefficients.

In this case, $\alpha_1$ is our estimate of the impact of using the GDS-15 to detect severe depression at baseline on depression severity six-months later.

Final model selection was based on minimising the Akaike's Information Criterion (AIC) and on the results of an F-test. Whilst AIC can demonstrate whether one model fits the data better than another, it does not give an indication of how well the model fits the data overall. For this reason, the F-test approach as described by Lee and Lemieux [31] was also conducted.

### Public patient involvement

There was no public or patient involvement for this analysis; however, the REFORM trial as a whole had a patient reference group, comprising of four people who were representative of trial participants, who gave comments on patient-facing material including the questionnaires containing the GDS-15 instrument.

## Results

In total, 895 (88.6%) of the 1010 participants randomised into REFORM had a valid baseline and six-month GDS-15 score and were included in this RDD analysis. Among these 895 participants, the mean GDS-15 baseline score was 3.5 (SD 3.0, median 3.0, range 0–15); 639 (71.4%) scored between 0 and 4, 204 (22.8%) between 5 and 9, and 52 (5.8%) between 10 and 15, indicating severe depression. These 52 participants were recruited from 43 GP surgeries (average 1.2 per surgery, range 1 to 3). At baseline the mean GDS score among those scoring <10 was 3.0 (SD 2.3, median 3) and among those scoring 10 or more was 11.5 (SD 1.4, median 11).

At six months, the overall mean GDS-15 score was 3.7 (SD 3.1, median 3.0, range 0–15); 618 (69.1%) scored between 0 and 4, 221 (24.7%) between 5 and 9, and 56 (6.3%) between 10 and 15. At 6 months the mean GDS score among those scoring <10 was 3.4 (SD 2.8, median 3) and among those scoring 10 or more was 9.4 (SD 2.6, median 9). Of the 52 participants with a score of 10 or more at baseline, 31 (59.6%) were no longer severely depressed at six months, of which 30 were still mildly depressed but one dropped to within the 'normal' range.

The correlation between the GDS-15 score at baseline and at six months was 0.76 (95% confidence interval (CI), 0.73 to 0.79)

### Manipulation of the continuous assignment variable

Visual inspection of a histogram of the baseline GDS-15 scores for the 895 participants with available data showed a positively skewed distribution and no clear evidence of manipulation around the cut-point of 10 (**Fig 1**).

### Baseline characteristics and testing continuity

Baseline characteristics of the participants by GDS-15 score, are presented in Table 1. There are some notable differences between the two groups. For example, the percentages of those with arthritis and those on four or more medications are higher in the group that scored 10 or more on the GDS-15 than those who scored less than 10.

Continuity was evaluated by comparing participants who scored slightly below the threshold with those who scored slightly above it (Table 2). The majority of characteristics were

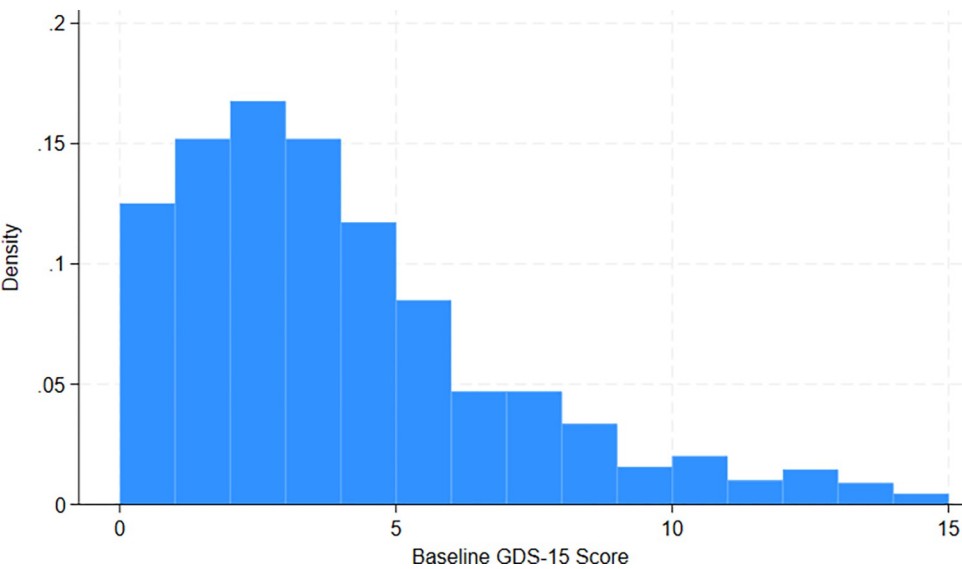

**Fig 1. Histogram of GDS-15 scores during baseline data collection.**

balanced and continuous around the continuous assignment variable cut point. There are a few exceptions such as the difference between groups in the number of participants taking four or more prescribed medications at baseline and EQ-5D-5L score. However, the validity of the regression discontinuity design still holds as we might expect one or two significant results just by chance given the number of comparisons tested.

## Regression discontinuity analysis results

Several functional forms of the regression model were estimated, starting with the simplest linear regression model, which adjusted only for the treatment (GP contacted) and GDS-15 score at baseline and increasing in complexity.

The linear regression model with no interaction term, and cubic regression model with an interaction term, produced the lowest AIC value (with and without covariates) (Table 3).

Again, starting with the simplest linear regression model, an F-test was calculated and was not found to be statistically significant (p = 0.40). This suggests that the simple linear regression model adequately depicts the relationship between the outcome and rating variables and therefore can serve as an appropriate choice for the RDD estimation model as opposed to a more complex model.

Fig 2 shows a scatter plot of participants' GDS-15 score at six months against their GDS-15 score at baseline with the fitted regression line from the selected model. There is a noticeable drop in the regression line at the cut-point, with those scoring 10 or higher on the GDS-15 at baseline scoring less on the GDS-15 at six months than expected, with a clear discontinuity present. The regression coefficient of interest is -1.08 (95% CI -1.83 to -0.33, p = 0.005), which indicates that participants who scored 10 or more and whose GP was subsequently contacted had a just over one-point reduction in depression severity six months later.

The robustness of the linear regression model and sensitivity of the treatment estimate was assessed by dropping the outermost 1% of the data and fitting the linear regression model again. When discarding the outermost 1%, the treatment estimate becomes -1.23 (SE 0.43, 95% CI -2.09 to -0.38), an absolute change of 0.15.

**Table 1. Participant baseline characteristics stratified by GDS-15 score.**

| Characteristic | | Baseline GDS-15 Score | | |
| --- | --- | --- | --- | --- |
| | | < 10 (Not Severely Depressed) N = 843 | ≥ 10 (Severely Depressed) N = 52 | Total N = 895 |
| Age (years), mean (SD) | | 77.7 (7.0) | 76.0 (7.3) | 77.6 (7.1) |
| Gender, n (%) | Male | 328 (38.9) | 25 (48.1) | 353 (39.4) |
| | Female | 515 (61.1) | 27 (51.9) | 542 (60.6) |
| Ethnicity, n (%) | White | 836 (99.2) | 51 (98.1) | 887 (99.1) |
| | Other | 3 (0.3) | 1 (1.9) | 4 (0.4) |
| REFORM Treatment | Usual Care | 435 (51.6) | 26 (50.0) | 461 (51.5) |
| Group, n (%) | Intervention | 408 (48.4) | 26 (50.0) | 434 (48.5) |
| Comorbidity†, n (%) | Arthritis | 477 (56.6) | 40 (76.9) | 517 (57.8) |
| | Depression | 61 (7.2) | 21 (40.4) | 82 (9.2%) |
| | Diabetes | 268 (31.8) | 26 (50.0) | 294 (32.8) |
| | Dizziness/Vertigo | 147 (17.4) | 22 (42.3) | 169 (18.9) |
| | Meniere's Disease | 25 (3.0) | 5 (9.6) | 30 (3.4) |
| | Numbness in feet | 132 (15.7) | 15 (28.8) | 147 (16.4) |
| | Osteoporosis | 124 (14.7) | 7 (13.5) | 131 (14.6) |
| Body Mass Index, mean (SD) | | 27.6 (5.1) | 30.4 (8.0) | 27.8 (5.4) |
| ≥4 Medications, n (%) | | 496 (58.8) | 45 (86.5) | 541 (60.4) |
| I tend to bounce back | Not true at all | 9 (1.1) | 2 (3.8) | 11 (1.2) |
| after illness or | Rarely true | 11 (1.3) | 8 (15.4) | 19 (2.1) |
| hardship, n(%) | Sometimes true | 115 (3.6) | 19 (36.5) | 134 (15.0) |
| | Often true | 248 (29.4) | 17 (32.7) | 265 (29.6) |
| | True nearly all of the time | 456 (54.1) | 5 (9.6) | 461 (51.5) |
| I am able to adapt to | Not true at all | 10 (1.2) | 2 (3.8) | 12 (1.3) |
| change, n (%) | Rarely true | 20 (2.4) | 5 (9.6) | 25 (2.8) |
| | Sometimes true | 162 (19.2) | 22 (42.3) | 184 (20.6) |
| | Often true | 255 (30.2) | 20 (38.5) | 275 (30.7) |
| | True nearly all of the time | 393 (46.6) | 2 (3.8) | 395 (44.1) |
| EQ-5D-5L score, mean (SD) | | 0.7 (0.2) | 0.4 (0.3) | 0.7 (0.2) |
| Post-16 Education, n (%) | | 485 (57.5) | 26 (50.0) | 511 (57.1) |
| Degree or Equivalent, n (%) | | 318 (37.7) | 15 (28.8) | 333 (37.2) |
| Living | Live alone | 361 (42.8) | 26 (50.0) | 387 (43.2) |
| Arrangements†, n (%) | Live with a partner | 433 (51.4) | 24 (46.2) | 457 (51.1) |
| | Live with a friend/relative | 44 (5.2) | 1 (1.9) | 45 (5.0) |
| | Sheltered accommodation | 25 (3.0) | 3 (5.8) | 27 (3.1) |

† Not mutually exclusive, participants asked to cross all that apply.

## Discussion

Depression among older people is a large source of morbidity in the community. There has been no large RCT to test the effectiveness of primary care case finding among the general population of older people in the UK. In this study we undertook a regression discontinuity analysis of people aged 65 years and above who were registered with local NHS podiatry services and had at least one risk factor for falling. As part of the REFORM trial we gave participants the GDS-15 to complete at baseline and six months. At baseline, 29% of participants had GDS-15 scores indicating they were at least mildly depressed, with 6% exhibiting symptoms of

**Table 2. Comparison of characteristics to test balance and continuity between participants who scored slightly above and below the continuous assignment variable threshold.**

| Characteristic | | Baseline GDS-15 Score | | |
|---|---|---|---|---|
| | | 8–9 N = 44 | 10–11 N = 27 | P-value |
| Age (years), mean (SD) | | 78.2 (7.7) | 77.4 (7.8) | 0.66 |
| Gender†, n (%) | Male | 21 (47.7) | 11 (40.7) | 0.57 |
| | Female | 23 (52.3) | 16 (59.3) | |
| Ethnicity†, n (%) | White | 44 (100.0) | 27 (100.0) | N/A |
| REFORM Treatment | Usual Care | 18 (40.9) | 12 (44.4) | 0.77 |
| Group†, n (%) | Intervention | 26 (59.1) | 15 (55.6) | |
| Comorbidity†, n (%) | Arthritis | 26 (59.1) | 20 (74.1) | 0.20 |
| | Depression | 10 (22.7) | 11 (40.7) | 0.11 |
| | Diabetes | 14 (31.8) | 12 (44.4) | 0.28 |
| | Dizziness/Vertigo | 16 (36.4) | 12 (44.4) | 0.50 |
| | Meniere's Disease | 2 (4.6) | 2 (7.4) | 0.61 |
| | Numbness in feet | 8 (18.2) | 9 (33.3) | 0.15 |
| | Osteoporosis | 9 (20.5) | 6 (22.2) | 0.86 |
| Body Mass Index, mean (SD) | | 29.9 (5.9) | 29.7 (8.7) | 0.91 |
| ≥4 Medications†, n (%) | | 30 (68.2) | 24 (88.9) | 0.05 |
| I tend to bounce back after illness or hardship†, n(%) | Not true at all | 0.0 (0.0) | 0.0 (0.0) | 0.33 |
| | Rarely true | 3 (6.8) | 5 (18.5) | |
| | Sometimes true | 15 (34.1) | 9 (33.3) | |
| | Often true | 17 (38.6) | 10 (37.0) | |
| | True nearly all of the time | 8 (18.2) | 2 (7.4) | |
| I am able to adapt to change†, n (%) | Not true at all | 2 (4.6) | 0 (0.0) | 0.12 |
| | Rarely true | 3 (6.8) | 4 (14.8) | |
| | Sometimes true | 21 (47.7) | 9 (33.3) | |
| | Often true | 11 (25.0) | 12 (44.4) | |
| | True nearly all of the time | 7 (15.9) | 1 (3.7) | |
| EQ-5D-5L score, mean (SD) | | 0.6 (0.2) | 0.4 (0.3) | 0.03 |
| Post-16 Education†, n (%) | | 22 (50.0) | 13 (48.1) | 0.87 |
| Degree or Equivalent†, n (%) | | 14 (31.8) | 10 (37.0) | 0.57 |
| Living Arrangements†, n (%) | Live alone | 20 (45.5) | 16 (59.3) | 0.26 |
| | Live with a partner | 21 (47.7) | 10 (37.0) | 0.38 |
| | Live with a friend/relative | 3 (6.8) | 0 (0.0) | 0.17 |
| | Sheltered accommodation | 2 (4.6) | 1 (3.7) | 0.86 |

† Chi-squared test used; t-test used for all other comparisons. Highlighted rows indicate statistical differences at the 5% significance level.

severe depression (score of 10 or more). These figures are in line with national findings that around a quarter of older people suffer from depression [2]. For ethical reasons we contacted the GPs of all participants who scored 10 points or above on the GDS-15 at enrolment to the REFORM trial. Following completion of the study, we realised that these data were appropriate to conduct a retrospective regression discontinuity analysis, and observed that, after six months, this intervention resulted in a significant improvement in the GDS-15 score, suggesting that a form of routine screening using the GDS-15 might be effective at reducing the severity of depression in older people.

These findings differ from those of a recent cluster RCT in the US to determine the effect of case-finding for depression, which was unable to demonstrate any benefit [33]. Thirteen

**Table 3. Different functional forms which were estimated, alongside the treatment estimate and AIC values.**

| Functional form | Treatment estimate | SE, 95% CI | p-value | AIC |
|---|---|---|---|---|
| *No covariates* | | | | |
| Linear | -1.08 | 0.38 (-1.83, -0.33) | 0.005 | 3785.7 |
| Linear interaction | -0.84 | 0.46 (-1.74, 0.08) | 0.072 | 3786.8 |
| Quadratic | -0.67 | 0.59 (-1.83, 0.48) | 0.255 | 3786.8 |
| Quadratic interaction | -0.52 | 0.63 (-1.76, 0.72) | 0.411 | 3790.1 |
| Cubic | -0.58 | 0.60 (-1.75, 0.60) | 0.334 | 3788.1 |
| Cubic interaction | 1.08 | 0.91 (-0.71, 2.87) | 0.237 | 3785.5 |
| *With covariates* | | | | |
| Linear | -1.02 | 0.38 (-1.77, -0.27) | 0.008 | 3772.2 |
| Linear interaction | -0.83 | 0.47 (-1.75, 0.08) | 0.074 | 3773.7 |
| Quadratic | -0.71 | 0.59 (-1.87, 0.45) | 0.230 | 3773.7 |
| Quadratic interaction | -0.54 | 0.64 (-1.80, 0.70) | 0.389 | 3777.0 |
| Cubic | -0.63 | 0.60 (-1.81, 0.55) | 0.294 | 3775.2 |
| Cubic interaction | 0.99 | 0.92 (-0.80, 2.79) | 0.277 | 3772.8 |

primary care medical clinics were randomised to intervention and control. As in our study, the GDS-15 was used to measure depressive symptoms and, for patients in the intervention group, their physician was informed if they scored over a certain threshold. The threshold was set at six, which was lower than our threshold, and different interventions were recommended to be offered to people scoring 6–10 and those scoring 11–15. There was no statistically significant difference in mean GDS-15 score at the two year follow-up.

## Strengths and limitations

There are a number of limitations to this study. First, all of our participants had podiatry needs and were at high risk of falling and so we cannot be sure our results will generalise to all people

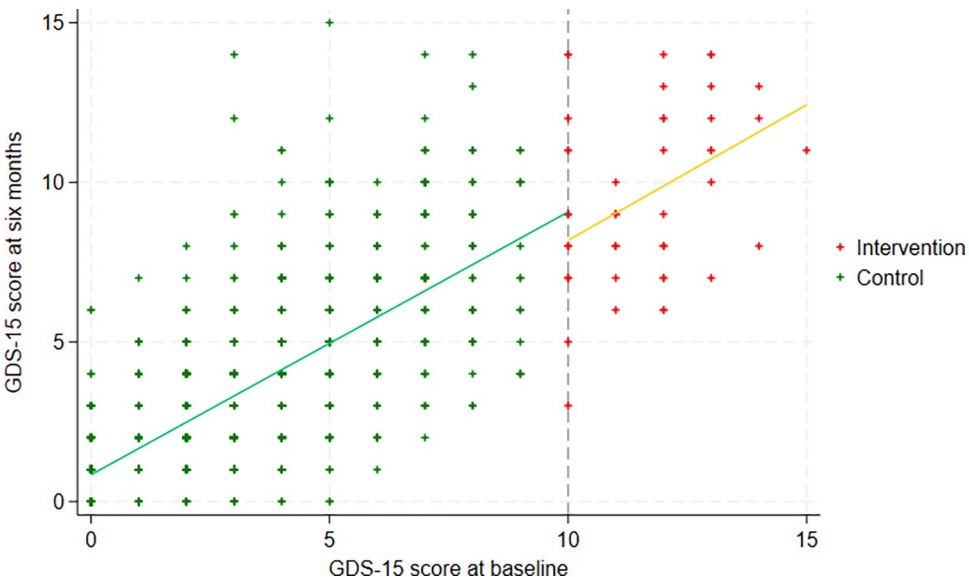

**Fig 2. Scatter plots of participant's GDS-15 score at six-months against their GDS-15 score at baseline, with the fitted values of the selected linear regression model.**

aged over 65 living in the community. Another limitation is that we are unsure exactly what, if any, intervention GPs gave to patients that led to an improvement in care, or what the time interval was between baseline and intervention, as these data were not collected. We assumed that GPs followed best professional practice, which may have included a consultation with the participant to discuss their symptoms and subsequent management of their depression with medication and/or psychological therapy, but in any future study it would be important to get a detailed description of what treatments were offered to (and accepted by) patients and how long this took.

Further there are limitations to the analysis design. The RDD is a quasi-experimental design and, in lacking random assignment as in an RCT, is not as powerful a design to investigate the effectiveness of an intervention. This was an opportunistic analysis, embedded within an RCT; therefore the sample size was constrained and not calculated specifically for this analysis. Generally, to answer the same research question, an RDD requires a much larger sample size than an RCT, at least three to five times more participants [34, 35]. Also, there is the possibility that the main trial REFORM intervention could also influence the GDS outcome here; however, in theory, within each group of the RDD (those who score <10 and those who score 10+), there will be an equal distribution of intervention and control participants for the main trial through randomisation. Therefore, any impact of the REFORM intervention will be equal across the RDD groups and so we can still investigate whether the GP intervention has an impact over and above any effect of the REFORM intervention. We adjusted for REFORM trial allocation in the analysis models to account for this. There was an approximately 1.1 point GDS-15 score reduction at the cut-point and whether this is clinically meaningful is uncertain; however, previous studies have utilised a similar (1.2 point) change in the GDS-15 as a clinically important difference [36]. The parametric approach to a regression discontinuity design requires the selection of a model that best fits the data. Different techniques to select the final model may give different results. Here, when we considered the lowest AIC, there was little difference between the linear regression and cubic regression with an interaction; however, these gave treatment effects that were qualitatively different, similar in magnitude but in the opposite direction. When the final model is specified, one sensitivity check that is suggested includes repeating the analysis with the outermost 1%, 5% and 10% data points excluded. However given the small sample size here and that only 6% of participants scored above the cut-point, we were restricted on the robustness checks that could be carried out. The treatment estimate when the outermost 1% of the data are dropped increased in magnitude, which may cast further doubt that the functional form (the simplest linear regression model) is correctly specified.

On the other hand, a strength of the regression discontinuity design is the way in which a treatment effect can be visualised. The discontinuity or drop in the regression line makes it clear if a treatment effect is or is not present. In addition, it allows the intervention to be provided to all those who most require it and thus, in some cases, provides a more ethically justifiable alternative to an RCT; some patients who would receive the intervention in an RDD would be denied it in an RCT due to random allocation.

Another strength of the study is the generalisability to a wide number of GP surgeries. The 52 participants with a GDS-15 baseline score of 10 or more were recruited from 43 GP surgeries, which suggests that the effect, if true, is not driven by the management strategies of a 'few' GPs and is likely to be representative of the impact of typical GP care.

In summary, a serendipitous evaluation suggests that routine case finding for depression with the GDS-15 may improve the mental wellbeing of older people. Given the promise indicated from such a simple intervention, the feasibility, acceptability, effectiveness, and cost-effectiveness of a well-designed case-finding approach is worthy of further exploration. In

addition, our exemplar demonstrates how a regression discontinuity analysis can be conducted retrospectively or prospectively embedded in other studies to answer a further research question, or at least to generate plausible future research hypotheses.

## Supporting information

**S1 Fig. Geriatric depression scale (15-items).**
(DOCX)

## Acknowledgments

We thank the REFORM trial participants for their participation in the REFORM trial and providing outcome data.

^Members of the 'REFORM trial' author group: Sarah Cockayne, Caroline Fairhurst, Joy Adamson, David Torgerson, Sara Rodgers, Arabella Scantlebury, Belen Corbacho Martin, Catherine Hewitt, Kate Hicks, Zoe Richardson, Judith Watson (York Trials Unit, Department of Health Sciences, University of York, York, UK), Wesley Vernon (University of Huddersfield, Huddersfield, UK), Lorraine Green, Anne Maree Keenan, Anthony C Redmond (Leeds Institute of Rheumatic and Musculoskeletal Medicine, Faculty of Medicine and Health, University of Leeds, Leeds, UK, and National Institute for Health Research (NIHR) Leeds Musculoskeletal Biomedical Research Unit, Leeds Teaching Hospitals NHS Trust, Leeds, UK), Robin Hull (Podiatry Department, Harrogate and District NHS Foundation Trust, Harrogate, UK), Sarah E Lamb (Oxford NIHR Biomedical Research Unit, Oxford, UK), Caroline McIntosh (Discipline of Podiatric Medicine, National University of Ireland Galway, Galway, Ireland), and Hylton B Menz (School of Allied Health, College of Science, Health and Engineering, La Trobe University, Melbourne, VIC, Australia).

Sarah Cockayne is the lead author of the REFORM trial group (sarah.cockayne@york.ac.uk).

## Author Contributions

**Conceptualization:** David J. Torgerson.

**Data curation:** Sarah Cockayne.

**Formal analysis:** Kalpita Baird, Caroline Fairhurst.

**Funding acquisition:** Sarah Cockayne, Caroline Fairhurst, Joy Adamson, Wesley Vernon, David J. Torgerson.

**Methodology:** Sarah Cockayne, David J. Torgerson.

**Project administration:** Sarah Cockayne, Caroline Fairhurst, Joy Adamson, David J. Torgerson.

**Writing – original draft:** Kalpita Baird, Ailish Byrne, Sarah Cockayne, Rachel Cunningham-Burley, Caroline Fairhurst, David J. Torgerson.

**Writing – review & editing:** Kalpita Baird, Ailish Byrne, Sarah Cockayne, Rachel Cunningham-Burley, Caroline Fairhurst, Joy Adamson, Wesley Vernon, David J. Torgerson.

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
