## [Decision Letter · Decision Letter 0]

27 Mar 2023

PONE-D-23-04252

Can routine assessment of older people’s mental health lead to improved outcomes: A regression discontinuity analysis

PLOS ONE

Dear Dr. Baird,

Thank you for submitting your manuscript to PLOS ONE. After careful consideration, we feel that it has merit but does not fully meet PLOS ONE’s publication criteria as it currently stands. Therefore, we invite you to submit a revised version of the manuscript that addresses the points raised during the review process.

We look forward to receiving your revised manuscript.

Kind regards,

Alex Schaefer, PhD

Associate Editor

PLOS ONE

Journal Requirements:

   "The author(s)s received no specific funding for this work. However, the original REFORM trial was funded by the National Institute for Health Research (NIHR) Health Technology Assessment (HTA) Programme (Programme grant number 09/77/01). "

   "I have read the journal's policy and the authors of this manuscript have the following competing interests: SC, CF, JA, WV, and DT received funding from the National Institute for Health Research (NIHR) Health Technology Assessment (HTA) paid to their employer, University of York, for the REFORM trial. All other authors declare no competing interests."

5. One of the noted authors is a group or consortium "REFORM trial". In addition to naming the author group, please list the individual authors and affiliations within this group in the acknowledgments section of your manuscript. Please also indicate clearly a lead author for this group along with a contact email address.

Additional Editor Comments:

The reviewers feel that the manuscript is well-written, but needs to undergo minor revisions. For example, they suggest that you state your hypothesis and/or research questions in the introduction section and provide a more comprehensive discussion of the importance/benefit of screening for depression in primary care settings. Please find the detailed comments appended below.

Reviewers' comments:

Reviewer's Responses to Questions

**Comments to the Author**

1. Is the manuscript technically sound, and do the data support the conclusions?

Reviewer #1: Yes

Reviewer #2: Yes

Reviewer #3: Yes

2. Has the statistical analysis been performed appropriately and rigorously? 

Reviewer #1: Yes

Reviewer #2: Yes

Reviewer #3: Yes

3. Have the authors made all data underlying the findings in their manuscript fully available?

Reviewer #1: Yes

Reviewer #2: No

Reviewer #3: No

4. Is the manuscript presented in an intelligible fashion and written in standard English?

Reviewer #1: Yes

Reviewer #2: Yes

Reviewer #3: Yes

5. Review Comments to the Author

Reviewer #1: Important note: This review pertains only to ‘statistical aspects’ of the study and so ‘clinical aspects’ [like medical importance, relevance of the study, ‘clinical significance and implication(s)’ of the whole study, etc.] are to be evaluated [should be assessed] separately/independently. Further please note that any ‘statistical review’ is generally done under the assumption that (such) study specific methodological [as well as execution] issues are perfectly taken care of by the investigator(s). This review is not an exception to that and so does not cover clinical aspects {however, seldom comments are made only if those issues are intimately / scientifically related & intermingle with ‘statistical aspects’ of the study}. Agreed that ‘statistical methods’ are used as just tools here, however, they are vital part of methodology [and so should be given due importance]. I look at the manuscript in/with statistical view point, other reviewer(s) look(s) at it with different angle so that in totality the review is very comprehensive. However, there should be efforts from authors side to improve (may be by taking clues from reviewer’s comments). Therefore, please do not limit the revision only (with respect) to comments made here.

COMMENTS: Although this manuscript is well drafted [and the study is excellent with respect to most of the aspects], I have few observations/concerns (different opinion) which are given below:

Firstly, {although on one hand I must congratulate and appreciate authors having used this design} it may please be noted that the RD (regression discontinuity) design lacks the random assignment feature. In addition to this, there are several issues* that disallow a status equal to the randomized experiment and the RD design has to be classified as a quasi-experimental design.

Few of the issues described in literature are:

- it is a new design and not yet as clear as the randomized experiment.

- the RD design heavily depends on correctly modeling the functional form of the relationship between the assignment variable and the outcome variable. Therefore, in order to obtain unbiased estimates, more sophisticated statistical analyses are required than is usual in experimental settings.

- the RD design has less power than the randomized experiment. This is mainly due to multicollinearity between the assignment and treatment variable. The loss of power is greater the more the cutoff deviates from the mean of the assignment variable. When nonlinear effects (e.g., interaction and quadratic effects) are present, multicollinearity increases as well as reduces the power of the design. In order to increase the power to 0.80, the RD design needs up to 3 times as many participants as a randomized experiment

Though sometimes the argument(s) made that “from a methodological point of view, inferences which are drawn from a well-implemented RD design are comparable in internal validity to conclusions from randomized experiments” it is not very convincing. It is often said that, in principle the RD design is as strong in internal validity as its randomized experimental alternatives, however, in practice, the validity of the RD design depends directly on how well the analyst can model the true pre-post relationship, certainly a nontrivial statistical problem.

The real allure of the RD Design is that it allows us to assign the treatment or program to those who most need or deserve it. Thus, the real attractiveness of the design is ethical – we don’t have to deny the program or treatment to participants who might need it as we do in randomized studies.

With this background information I have, few questions are raised in mind [though they may appear to be silly on prima-facie, you need to clarify these things because mind you that this is a scientific/academic document and so all details should be clearly/correctly communicated (do not take readers’ for granted)].

• As per line 26 [Design] data are from a randomised controlled trial. My question is {though in lines 142-143 it is stated that adjustment for covariates included that for participant’s REFORM trial allocation}, participant’s those who are randomly allocated to intervention arm of REFORM trial are already influenced/affected by that intervention [those who are randomly allocated to control arm (even if active control or placebo) of REFORM trial are already influenced/affected differently] then will it likely to affect your outcome.

• There may be substantial period between ‘baseline’ reading time and your intervention. What about that?

Section ‘Study Design and Statistical Methods’ in lines 106 onwards is fairly well drafted but authors may think including above (and/or other such) points. I agree with the ‘Limitations’ part highlighted/enumerated in lines 260 onwards [but not so much with ‘Strengths’ part of the section].

Except these minor points, the article is acceptable. However, mind you that as pointed out in ‘important note’ above “This review pertains only to ‘statistical aspects’ of the study and so ‘clinical aspects’ should be assessed separately/independently.

Reviewer #2: The study is well structured and described in the manuscript, and I think it brings important and insightful contributions regarding ways to improve mental health assistance to the older population. However, I made some suggestions and recommendations detailed in the manuscript's comment boxes, attached to this review. Basically, I suggest the authors (1) to state their hypothesis and/or research questions in the introduction; (2) to justify why they chose to run the sharp version of RDD; (3) to describe other instruments and questionnaires used along with the GDS-15 to collect data, especially those described in the results; (4) to revise Table 1, there are many points to pay attention. I recommend running statistical analyses to check for significant differences between groups in the results in Table 1. There is no information in methods about the instruments displayed in Table 1, so the reader can't interpret appropriately the results. I recommend you remove this data from the table or describe the instruments in the methods and how we interpret them. The sum of the cases for gender in Table 1 needs to be revised, it does not match. Please, specify what BMI means; (5) In Table 2 you need to highlight the cells of "during the past 4 weeks, have you worried about a fall" to indicate there was a significant difference between groups; (6) The results must be interpreted taking into account the sociodemographic background of the sample, and I think that authors should go deeper in this matter in the discussion

Reviewer #3: I found the paper to be very interesting. The authors adequately described and analysed their findings. While the effect of the intervention (1.1-point reduction) may not be clinically significant, the authors have provided appropriate context and reported on the uncertainty of it in the discussion section. The paper is somewhat concise, it is well written. I do have a few suggestions for the authors:

(Introduction) It would be helpful to provide a more comprehensive discussion on the importance/benefits of screening for depression in primary care settings.

(Methods/Results) It is not clear when the data were collected, which could be clarified in the methods or results section.

(Methods) It would be informative to include data from the second follow-up assessment (12 months) to evaluate the long-term effects of the intervention.

(Results) It would be helpful to report means and medians of GDS-15 scores in both groups (<10/10+) at baseline and 6 months to provide readers with a better understanding of the true effect of the intervention.

(Results) Including information about depression treatment, if available, would be useful. If such information is not available, the authors should acknowledge this as a limitation of their study.

6. PLOS authors have the option to publish the peer review history of their article (what does this mean?). If published, this will include your full peer review and any attached files.

Reviewer #1: No

Reviewer #2: **Yes: **Heloisa Goncalves Ferreira

Reviewer #3: No

---

## [Author Response · Author response to Decision Letter 0]

20 Oct 2023

Dear Sir/Madam,

We previously submitted our manuscript titled “Can routine assessment of older people’s mental health lead to improved outcomes: A regression discontinuity analysis” [PONE-D-23-04252]. We thank the Academic Editor and reviewers for their comments and feedback and have addressed each of the points raised in turn below, as well as in the manuscript. 

Response to Reviewers

Rebuttal Letter

Comments raised by the Academic Editor 

Formatting Requirements

Please ensure that your manuscript meets PLOS ONE’s style requirements, including those for file naming. The PLOS ONE style templates can be found at https://journals.plos.org/plosone/s/file?id=wjVg/PLOSOne_formatting_sample_main_body.pdf and https://journals.plos.org/plosone/s/file?id=ba62/PLOSOne_formatting_sample_title_authors_affiliations.pdf

Thank you, we have made sure the formatting of the manuscript complies with these templates.

Role of Funder Statement 

Our previous financial disclosure statement read as follows: “The author(s) received no specific funding for this work. However, the original REFORM trial was funded by the National Institute for Health Research (NIHR) Health Technology Assessment (HTA) Programme (Programme grant number 09/77/01).” 

To this should be added the following: “The funders had no role in study design, data collection, and analysis, decision to publish, or preparation of the manuscript.”

Competing Interests Statement 

Our previous Competing Interests statement read as follows: “I have read the journals’ policy and the authors of this manuscript have the following competing interests: SC, CF, JA, WV and DT received funding from the National Institute for Health Research (NIHR) Health Technology Assessment (HTA) paid to their employer, University of York, for the REFORM trial. All other authors declare no competing interests.”

This has now been amended to: “I have read the journal’s policy and the authors of this manuscript have the following competing interests: SC, CF, JA, WV and DT received funding from the National Institute for Health and Care Research (NIHR, previously just called the National Institute for Health Research) Health Technology Assessment (HTA) paid to their employer, University of York, for the REFORM trial. All other authors declare no competing interests. This does not alter our adherence to PLOS ONE policies on sharing data and materials.”

Data Availability Statement 

We note that you have indicated that data from this study are available upon request. PLOS only allows data to be available upon request if there are legal or ethical restrictions on sharing data publicly.

For this study there are ethical restrictions on sharing a de-identified data set. Consent to share anonymous data from the study was not obtained from participants. It is therefore not possible to openly share the data with other researchers via a public repository, and data are only available upon request. Data requests are to be sent to dohs-stats@york.ac.uk.

Acknowledgements

One of the noted authors is a group or consortium “REFORM trial’. In addition to naming the author group, please list the individual authors and affiliations within this group in the acknowledgements section of your manuscript. Please also indicate clearly a lead author for this group along with a contact email address.

Thank you, we have added the extra authors from the REFORM trial group (who are not named authors of this manuscript) to the Acknowledgements and indicated who the lead author for this group is. 

Ethics Statement

Please include your full ethics statement in the ‘methods’ section of your manuscript file. In your statement, please include the full name of the IRB or ethics committee who approved or waived your study, as well as whether or not you obtained informed written or verbal consent. If consent was waived for your study, please include this information in your statement as well.

We have added the following statement to the ‘methods’ section of the manuscript:

Regulatory approval for the study was obtained from the East of England – Cambridge East Research Ethics Committee (REC) on 9 November 2011 (REC reference number 11/EE/0379). Galway REC approved the study on 26 April 2013 (REC reference number C.A 886). The University of York, Department of Health Sciences Research Governance Committee approved the study on 2 August 2011. Research management and governance approval was obtained for each trust. Participation in the study was voluntary, and all participants gave written informed consent before taking part in the REFORM study. 

Supporting Information Files 

Please indicate captions for your Supporting Information files at the end of your manuscript, and update any in-text citations to match accordingly. http://journals.plos.org/plosone/s/supporting-information.

We have indicated captions for this Supporting Information file at the end of the manuscript and updated the in-text citation from Appendix A to S1 Fig.

References 

We have noticed that in our previous manuscript the following reference contained a broken link. This link has now been updated in our manuscript. 

 Age, U., Hidden in plain sight: The unmet mental health needs of older people. Age UK, London, available at: www. ageuk. org. uk/brandpartnerglobal/wiganboroughvpp/hidden_in_plain_sight_older_ peoples_mental_health. pdf (accessed 19 February 2019), 2016.

We have noticed that since our earlier submission the guidelines in the following reference have been updated and have updated the reference in our manuscript to reflect the most recent guidelines.

Excellence, N.I.f.C., Depression in adults: Recognition and management. Clinical guideline [CG90]. 2009, NICE London.

We have also included new references in our introduction in response to the feedback from the reviewers. These are as follows:

10. Thombs, B.D., et al., Rethinking recommendations for screening for depression in primary care. Canadian Medical Association Journal, 2012. 184(4): p. 413-418.

11. Kessler, D., D. Sharp, and G. Lewis, Screening for depression in primary care. British Journal of General Practice, 2005. 55(518): p. 659-660.

12. Thombs, B.D. and R.C. Ziegelstein, Does depression screening improve depression outcomes in primary care? BMJ : British Medical Journal, 2014. 348: p. g1253.

Final comments raised by the Academic Editor 

The reviewers feel that the manuscript is well-written but needs to undergo minor revisions. For example, they suggest that you state your hypothesis and/or research question in the introduction section and provide a more comprehensive discussion of the importance/benefit of screening for depression in primary care settings. 

Thank you, we have responded to each of the reviewer’s comments in turn below.

Comments raised by Reviewer #1

This review pertains only to ‘statistical aspects’ of the study and so ‘clinical aspects’ [like medical importance, relevance of the study, clinical significance and implication(s) of the whole study, etc] are to evaluated [should be assessed] separately/independently. 

Further please note that any ‘statistical review’ is generally done under the assumption that (such) study specific methodological [as well as execution] issues are perfectly taken care of by the investigator(s). This review is not an exception to that and so does not cover clinical aspects [however, seldom comments are made only if those issues are intimately/scientifically related & intermingle with ‘statistical aspects’ of the study]

Agreed that ‘statistical methods’ are used as just tools here, however, they are vital part of methodology [and so should be given due importance]. I look at the manuscript in/with statistical view point, other reviewer(s) look(s) at it with different angle so that in totality the review if very comprehensive. However, there should be efforts from the authors side to improve (maybe by taking clues from reviewer’s comments). Therefore, please do not limit the revision only (with respect) to comments made here. 

Although this manuscript is well drafted [and the study is excellent with respect to most of the aspects], I have a few observations/concerns (different opinion) which are given below:

Firstly, (although on one hand I must congratulate and appreciate authors having used this design) it may please be noted that the RD (regression discontinuity) design lacks the random assignment feature. In addition to this, there are several issues that disallow a status equal to the randomised experiment and the RD design has to be classified as a quasi-experimental design.

Few of the issues described in literature are:

• It is a new design and not yet as clear as the randomized experiment

• The RD design heavily depends on correctly modeling the functional form of the relationship between the assignment variable and the outcome variable. Therefore, in order to obtain unbiased estimates, more sophisticated statistical analyses are required than is usual in experimental settings.

• The RD design has less power than the randomized experiment. This is mainly due to multicollinearity between the assignment and treatment variable. The loss of power is greater the more the cutoff deviates from the mean of the assignment variable. When nonlinear effects (e.g. interaction and quadratic effects) are present, multicollinearity increases as well as reduces the power of the design. In order to increase the power to 0.80, the RD design needs up to 3 times as many participants as a randomised experiment

Though sometimes the argument(s) made that “from a methodological point of view, inferences which are drawn from a well-implemented RD design are comparable in internal validity to conclusions from randomised experiments” it is not very convincing. It is often said that, in principle the RD design is as strong in internal validity as its randomised experimental alternatives, however, in practice, the validity of the RDD depends directly on how well the analyst can model the true pre-post relationship, certainly a non-trivial statistical problem.

We acknowledge these limitations of the RDD relative to an RCT and have added the following to the Discussion:

“The RDD is a quasi-experimental design and, in lacking random assignment as in an RCT, is not as powerful a design to investigate the effectiveness of an intervention.”

We already acknowledged that the RDD relies on specifying the correct functional form for the data, and that a greater sample size is required than for an RCT.

The real allure of the RDD is that it allows us to assign the treatment or program to those who most need or deserve it. Thus, the real attractiveness of the design is ethical - we don’t have to deny the program or treatment to participants who might need it as we do in randomized studies.

Thank you, we agree this is a strength of the RDD over an RCT and have added the following to the Discussion:

“In addition, it allows the intervention to be provided to all those who most require it and thus, in some cases, provides a more ethically justifiable alternative to an RCT; some patients who would receive the intervention in an RDD would be denied it in an RCT due to random allocation.”

With this background information I have, few questions are raised in mind [though they may appear to be silly on prima-facie, you need to clarify these things because mind you that this is a scientific/academic document and so all details should be clearly/correctly communicated (do not take readers’ for granted)]

As per line 26 [Design] data are from a randomised controlled trial. My question is {though in lines 142-143 it is stated that adjustment for covariates included that for participant’s REFORM trial allocation}, participant’s those who are randomly allocated to intervention arm of REFORM trial are already influenced/affected by that intervention [those who are randomly allocated to control arm (even if active control or placebo) of REFORM trial are already influenced/affected differently] then will it likely to affect your outcome.

We agree but, in theory, within each group of the RDD (those who score <10 and those who score 10+), there will be an equal distribution of intervention and control participants for the main trial. Therefore, any impact of the main trial intervention will be equal across the RDD groups and so we can still investigate whether the GP intervention has an impact over and above any effect of the REFORM intervention. As the researcher points out, we did specifically include main trial allocation as a covariate in the analysis models to adjust for this effect.

We have added this point to the Discussion.

There may be substantial period between ‘baseline’ reading time and your intervention. What about that?

We acknowledge this is a limitation and have added to the Discussion:

“Another limitation is that we are unsure exactly what, if any, intervention GPs gave to patients that led to an improvement in care, or what the time interval was between baseline and intervention, as these data were not collected. We assumed that GPs followed best professional practice, which may have included a consultation with the participant to discuss their symptoms and subsequent management of their depression with medication and/or psychological therapy, but in any future study it would be important to get a detailed description of what treatments were offered to (and accepted by) patients and how long this took. ”

Section ‘Study Design and Statistical Methods’ in lines 106 onwards is fairly well drafted but authors may think including above (and/or other such) points. I agree with the ‘limitations’ part highlighted/enumerated in lines 260 onwards [ but not so much the ‘Strengths’ part of the section’]

Please see responses above. 

Except these minor points, the article is acceptable. However, mind you that as pointed out above ‘This review pertains only to statistical aspects’ of the study and so clinical aspects should be assessed separately/independently.

We thank the reviewer for their helpful comments. 

Comments raised by Reviewer #2

The study is well structured and described in the manuscript, and I think it brings important and insightful contributions regarding ways to improve mental health assistance to the older population. However, I made some suggestions and recommendations detailed in the manuscript’s comment boxed, attached to this review. Basically, I suggest the authors:

To state their hypothesis and/or research questions in the introduction

Our research question was whether case finding of depression in older adults reduces depressive symptoms six months later. This is stated at the end of the introduction:

“In this paper we report a natural experiment using a regression discontinuity analysis to assess whether case finding of depression in older adults reduces depressive symptoms six months later.” 

To justify why they chose to run the sharp version of the RDD

We ran the sharp version of the RDD as all participants bar one received the intervention. The choice of the sharp version of the RDD reflects that the intervention was deterministic, rather than probabilistic in nature. We did not feel it was necessary to run a fuzzy version of the RDD just as one patient did not receive the intervention.

To describe other instruments and questionnaires used along with the GDS-15 to collect data, especially those described in the results

Upon reflection, we have decided to streamline Tables 1 and 2, by removing data on the Short Falls Efficacy Scale, Frenchay Activities Index, referral to a falls clinic, data on falls in the last six months, worries over having a fall over the last four weeks, broken bones in the last 12 months, current use of an insole and orthotic and modifications to shoes as these outcomes were more pertinent to the main REFORM trial. 

To revise Table 1, there are many points to pay attention. I recommend running statistical analyses to check for significant differences between groups in the results in Table 1.

We visually checked for notable differences between the groups presented in Table 1 and have commented on these. We have used statistical tests to check for significant differences between groups around the cutpoint in Table 2 as this is a recommended part of an RDD analysis.

B. There is no information in the methods about the instruments displayed in Table 1, so the reader can’t interpret appropriately the results. I recommend you remove this data from the table or describe the instruments in the methods and how we interpret them.

See previous comment and response 

C. The sum of the cases for gender in Table 1 needs to be revised, it does not match.

This has been revised now.

D. Please specify what BMI means

Changed to Body Mass Index 

In Table 2 you need to highlight the cells of ‘during the past 4 weeks, have you worried about a fall” to indicate there was a significant difference between groups

The p-value for this variable is 0.09 and therefore is not significant at the 5% level and remains unhighlighted in Table 2. We have reiterated this in the footnote to Table 2.

The results must be interpreted taking into account the sociodemographic background of the sample and I think that authors should go deeper in this matter in the discussion

In the original REFORM trial very little sociodemographic data were collected. Whilst it would have been possible to have used postcode to calculate indices of deprivation, we no longer hold this data, as the standard operating procedures at the point of when the trial was set up, allowed for storage of personal data for five years. This time point has now passed and data relating to postcode have therefore been destroyed.

All other data collected from/about the participants is summarised in the Tables provided. 

Comments raised by Reviewer #3

I found the paper to be very interesting. The authors adequately described and analysed their findings. Whilst the effect of the intervention (-1.1 point reduction) may not be clinically significant , the authors have provided appropriate context and reported on the uncertainty of it in the discussion section. The paper is somewhat concise, it is well written. I do have a few suggestions for the authors:

(Introduction) It would be helpful to provide a more comprehensive discussion on the importance/benefits of screening for depression in primary care settings

Literature suggests that screening for depression in primary care settings is a highly debatable issue, with limited evidence on the importance and/or benefits of screening for depression in primary care settings. This has been amended and in our introduction, we now say:

The benefits of case finding for depression within primary care settings is widely debated. There is limited evidence to suggest that patients screened for depression had better outcomes than patients who were not, when the same treatment resources were available to both groups. Whilst screening could potentially identify people who do not recognise that they are experiencing depressive symptoms, leading to earlier detection, currently there is no strong evidence for routine case finding of depression among the older population. In this paper we report a natural experiment using a regression discontinuity analysis to assess whether case finding of depression in older adults reduces depressive symptoms six months later. 

(Methods/results) it is not clear when the data were collected, which could be clarified in the methods or results section

Thank you, we have clarified that baseline data were collected between November 2012 and November 2014.

(Methods) it would be informative to include data from the second follow-up assessment (12-months) to evaluate the long-term effects of the intervention

After consideration, we have not included the 12 month GDS scores as these data would be difficult to interpret. This is because we repeated the intervention at 6 months. Therefore, for any participant who scored over the cutpoint at 6 months who had scored below it at baseline, their GP was contacted. Therefore, a proportion of the original ‘control’ group at baseline would have received the intervention 6 months later, which would impact the long-term effect of the original intervention.

(Results) it would be helpful to report means and medians of GDS-15 scores in both groups (<10/10+) at baseline and 6 months to provide readers with a better understanding of the true effect of the intervention.

Thank you, we have added these statistics to the Results section of the manuscript. 

(Results) including information about depression treatment, if available, would be useful. If such information is not available, the authors should acknowledge this as a limitation of their study.

Data on the action/treatment undertaken in response to the GP receiving our letter were not collected in the REFORM trial as this was not an objective or outcome of the main trial. We already acknowledge this as a limitation in the Discussion, but have made this explicit in the Methods section now:

“We did not follow up with the patients or GPs regarding what, if any, action or treatment was undertaken as a result of the GP receiving our letter, as this was not related to any objectives of the main REFORM trial.”

In the Discussion we now say:

“Another limitation is that we are unsure exactly what, if any, intervention GPs gave to patients that led to an improvement in care, or what the time interval was between baseline and intervention, as these data were not collected. We assumed that GPs followed best professional practice, which may have included a consultation with the participant to discuss their symptoms and subsequent management of their depression with medication and/or psychological therapy, but in any future study it would be important to get a detailed description of what treatments were offered to (and accepted by) patients and how long this took.“

We resubmit our manuscript (within the six months of the decision date) having addressed these comments and very much look forward to receiving your feedback.

Kind regards,

Kalpita Baird

---

## [Decision Letter · Decision Letter 1]

1 Mar 2024

Can routine assessment of older people’s mental health lead to improved outcomes: A regression discontinuity analysis

PONE-D-23-04252R1

Dear Dr. Baird,

We’re pleased to inform you that your manuscript has been judged scientifically suitable for publication and will be formally accepted for publication once it meets all outstanding technical requirements.

Kind regards,

Vanessa Carels

Staff Editor

PLOS ONE

Additional Editor Comments (optional):

Reviewers' comments:

Reviewer's Responses to Questions

**Comments to the Author**

1. If the authors have adequately addressed your comments raised in a previous round of review and you feel that this manuscript is now acceptable for publication, you may indicate that here to bypass the “Comments to the Author” section, enter your conflict of interest statement in the “Confidential to Editor” section, and submit your "Accept" recommendation.

Reviewer #1: All comments have been addressed

2. Is the manuscript technically sound, and do the data support the conclusions?

Reviewer #1: (No Response)

3. Has the statistical analysis been performed appropriately and rigorously? 

Reviewer #1: (No Response)

4. Have the authors made all data underlying the findings in their manuscript fully available?

Reviewer #1: (No Response)

5. Is the manuscript presented in an intelligible fashion and written in standard English?

Reviewer #1: (No Response)

6. Review Comments to the Author

Reviewer #1: COMMENTS: All the comments are answered and most positively attended [I appreciate that many desired changes are made, especially the design details added]. I recommend the acceptance because the manuscript has now achieved the acceptable level (even earlier also I said: except highlighted minor points, the article is acceptable} in my opinion.

7. PLOS authors have the option to publish the peer review history of their article (what does this mean?). If published, this will include your full peer review and any attached files.

Reviewer #1: No

---

## [Editor Report · Acceptance letter]

8 Mar 2024

PONE-D-23-04252R1 

PLOS ONE

Dear Dr. Baird, 

I'm pleased to inform you that your manuscript has been deemed suitable for publication in PLOS ONE. Congratulations! Your manuscript is now being handed over to our production team.

Kind regards, 

on behalf of

Dr. Vanessa Carels 

Staff Editor

PLOS ONE